# Dielectrophoresis-Based SERS Sensors for the Detection of Cancer Cells in Microfluidic Chips

**DOI:** 10.3390/bios12090681

**Published:** 2022-08-25

**Authors:** Tomasz R. Szymborski, Marta Czaplicka, Ariadna B. Nowicka, Joanna Trzcińska-Danielewicz, Agnieszka Girstun, Agnieszka Kamińska

**Affiliations:** 1Institute of Physical Chemistry, Polish Academy of Sciences, Kasprzaka 44/52, 01-224 Warsaw, Poland; 2Faculty of Materials Engineering and Technical Physics, Poznan University of Technology, Piotrowo 3, 60-965 Poznan, Poland; 3Department of Molecular Biology, Institute of Biochemistry, Faculty of Biology, University of Warsaw, Miecznikowa 1, 02-096 Warsaw, Poland

**Keywords:** surface-enhanced Raman spectroscopy (SERS), dielectrophoresis, cancer cells, microfluidics, MCF-7, MDA-MB-231

## Abstract

The detection of freely circulating cancer cells (CTCs) is one of the greatest challenges of modern medical diagnostics. For several years, there has been increased attention on the use of surface-enhanced Raman spectroscopy (SERS) for the detection of CTCs. SERS is a non-destructive, accurate and precise technique, and the use of special SERS platforms even enables the amplification of weak signals from biological objects. In the current study, we demonstrate the unique arrangement of the SERS technique combined with the deposition of CTCs cells on the surface of the SERS platform via a dielectrophoretic effect. The appropriate frequencies of an alternating electric field and a selected shape of the electric field can result in the efficient deposition of CTCs on the SERS platform. The geometry of the microfluidic chip, the type of the cancer cells and the positive dielectrophoretic phenomenon resulted in the trapping of CTCs on the surface of the SERS platform. We presented results for two type of breast cancer cells, MCF-7 and MDA-MB-231, deposited from the 0.1 PBS solution. The limit of detection (LOD) is 20 cells/mL, which reflects the clinical potential and usefulness of the developed approach. We also provide a proof-of-concept for these CTCs deposited on the SERS platform from blood plasma.

## 1. Introduction

In recent years, much attention has been paid to developing effective methods to separate, isolate, capture and identify rare cell types, including circulating tumor cells (CTCs). CTCs are cancer cells that have been separated from the primary (or metastatic) tumor and circulate in the bloodstream [1]. The detection and monitoring of CTCs is important in many medical and clinical studies; most importantly, it is crucial for cancer screening, diagnosis, monitoring of treatment response and prognosis [1,2,3]. The CTC’s analysis of peripheral blood, known as ‘liquid biopsy’, has received special attention because it is less invasive, less expensive than tissue biopsy, and results are obtained quicker compared to traditional tissue biopsies, which take about a month to result. The limitation of this method is the small number of circulating tumor cells, which, even in metastatic cancer, is only 1 to 100 in 1 mL of blood [4]. Therefore, the detection and isolation of CTCs is a difficult task and requires long-term methods. CTC analysis methods rely on the isolation and detection stage, which is a long process that requires special and expensive technologies. The current CTCs isolation methods are based on the following: (i) physical separation taking into account the density, (ii) biochemical separation using antibodies against hematopoietic cells or the depletion of leukocytes and erythrocytes, (iii) separation using specific markers and markers modified with magnetic beads, and (iv) separation on the basis of the properties of matter, charge, migration properties or deformability [5,6].

Some CTCs isolation technologies have satisfactory performance, but they still have a few limitations. The main disadvantage is their low specificity for purity, viability and recovery rates as a result of the size overlap between the CTCs and normal blood cells [7]. For this reason, subsequent screening and detection processes are required. The most common detection method currently in use is the polymerase chain reaction (PCR), reverse transcription PCR (RT-PCR), scanning fluorescence microscopy, flow cytometry analysis based on monoclonal antibodies, laser scanning cytometry and immunofluorescence tests [8,9,10,11,12]. In recent years, the focus has been on the detection of CTCs using microfluidic reactions [13]. The capture of CTCs in the flow allows for the concentration of CTCs and their efficient analysis. There are many methods for isolating CTCs from blood samples in a microfluidic system, such as physical separation based on the fact that CTCs are larger than other blood components [14,15,16], dielectrophoresis [17,18,19,20,21,22,23], pinch flow [24,25,26] or ultrasonic resonances [27,28].

Park et al. [29] showed the efficient isolation of CTCs using a microfluidic system. However, this method relied on immunoaffinity and requires a thiol ligand exchange reaction with gold nanoparticles and adsorption of antibodies. Moreover, in this technique, the identification of CTCs was based on the immunofluorescence staining method. The entire detection process is time-consuming, laborious and requires expensive reagents. Additionally, problems with non-specific adsorption of antibodies and false positive results may occur.

In the current study, we used the phenomenon of dielectrophoresis (DEP), described for the first time by Pohl in the 1950s [30], which is a technique for manipulating bacteria and cancer cells in the microscale, and combined it with the SERS technique. Dielectrophoresis is a universal, label-free technique for detecting, analyzing, separating, fractionating or concentrating biological materials, e.g., bacteria, viruses, cancer cells, yeast and other [31,32,33,34,35].

The DEP is a phenomenon where cells, after the application of inhomogeneous alternating electric field (AC), can be attracted to the strong region of the electrical field gradient if they are more polarizable than the surrounding medium (positive DEP). They can also be repelled to the weak region of the electrical field gradient if particles are less polarizable than the surrounding medium (negative DEP) [36]. To manipulate particles via DEP differences in their size, geometry and dielectric properties (conductivity and electric permeability), both the particle and surrounding matrix can be utilized [37]. This mechanism and plethora of possible parameters lead to the use of DEP in many applications. For the first time, Pohl et al. successfully used the DEP phenomenon for the segregation of the living and dead cells of yeast [38]. DEP can be used to capture pathogens in a mixture [39], assess the resistance of neoplastic cells relative to selected drugs [40], or separate prostate and colon cancer [41], breast [42] or catch circulating tumor cells from the blood plasma of patients [36,43].

Various cell properties (size, morphology, deformability, mechanical, electrical or magnetic properties) can be a parameter for DEP to isolate cancer cells from blood [44,45,46]. Since many cells have distinctive size and shape characteristics, cell-size-based separation techniques are also typically preferred for classifying several types of cells [47,48,49,50]. Microfluidic techniques are also used together with dielectrophoresis: a DEP-based microfluidic device with continuous-flow separation of platelets from other blood cells due to their size difference was demonstrated [51,52]. Cancer cells and erythrocytes can be distinguished using DEP in microchamber device [53], where the non-uniform electric field was generated by the arrangement of alternating polarity electrodes. Another microfluidic device used DEP trapping for the separation of CTCs (MDA-MB-231) from the RBC sample [54]. Also, using a DEP-based microfluidic chip with optically transparent electrodes, MCF-7 CTCs were successfully separated from HCT116 CTCs [55].

Herein, we demonstrate a combination of microfluidics and the SERS detection system for selected breast-cancer cell studies. Due to the low concentration of CTCs in the blood and to minimalize the total number of steps in the analysis, we used microfluidic techniques (Lab-on-a-Chip) and dielectrophoresis to manipulate CTCs. Microfluidic techniques allow the handling of fluids at the microscale in addition to flow control, mixing and separation [56,57,58,59,60]. In the presented studies, we applied the microfluidic technique coupled with dielectrophoresis to separate CTCs and to deposit them on the custom-made SERS platform. The system, combined with advanced chemometric methods, will provide a fully functional tool for the detection and identification of cancer cells.

## 2. Materials and Methods

### 2.1. Cultivation and Preparation of Cancer Cells

Human breast adenocarcinoma cell lines, MCF-7 and MDA-MB-231, were used for experiments. Both cell lines came from the European Collection of Cell Cultures (ECACC, Salisbury, UK) and were supplied by Sigma-Aldrich (St. Louis, MO, USA). MCF-7 cells were cultured in a Minimum Essential Medium Eagle (MEM) supplemented with 2 mM glutamine and 1% MEM Non Essential Amino Acids Solution (100×). The MDA-MB-231 cells were cultured in Dulbecco’s Modified Eagle’s Medium (DMEM). Both media were additionally supplemented with 10% FBS, streptomycin (100 μg/mL) and penicillin (100 U/mL). The cell cultures were cultivated at 37 °C in a humidified atmosphere of 5% CO_2_. After reaching 80% of confluence, the cells were washed with a PBS buffer and detached from vessels with Accutase^®^ solution. Subsequently, the cells were collected, centrifuged at 250× *g* for 5 min at room temperature, washed with PBS and centrifuged again. The initial concentration (after the cultivation step) of cancer cells in PBS was ca. 10^6^ cells/mL After the final centrifugation, MCF-7 and MDA-MB-231 cells were resuspended in the 0.1 PBS or human blood plasma. The used volume of liquid to fill the DEP-chamber was typically 20 μL and the concentration of the CTCs was ca. 20 × 10^3^ for both: 0.1 PBS and blood plasma sample. For the sensitivity study, MDA-MB-231 cancer cells suspension (1 mL, 10^4^ cells/mL) was further diluted to various concentrations from 8 to 1400 cells in 1 mL of 0.1 PBS. The resultant cell samples were detected by the developed SERS-DEP method.

The electrical conductivity of the 0.1 PBS was monitored using a conductivity meter (VWR pHenomenal, model MU 6100H) with the proper CO11 conductivity probe. The measured electrical conductivity of the medium (σ_m_) was 0.19 S/m.

### 2.2. Microfluidic System for Handling of Cancer Cells

The microfluidic system that we have used for dielectrophoretic separation of the CTCs consists of three parts: cover plate, bottom plate and top plate (see Figure 1 and Figure 2). In the microfluidic chip, there is a microfluidic channel (0.6 mm × 0.6 mm) running along the x-axis of the chip. At the very center of the chip, the microfluidic chamber (diameter: 6 mm and depth: 0.7 mm) and chamber for the DEP counter-electrode (diameter: 15.0 mm and depth: 4.1 mm) were micromachined. The microfluidic chamber (machined in bottom plate) consists of a 450 μm mm cavity where the SERS platform (3 mm × 3 mm and ca. 500 μm in thickness) is placed (see Figure 1 and Appendix A in ESI). The SERS platform is located under the level of the channel; thus, the fluid with CTCs can fill the chamber and excess fluid can flow out of the chamber. In the top plate, the chamber for the DEP counter-electrode is located. The size of the chamber is machined with high accuracy relative to the diameter of the metal counter-electrode. 

After the assembly of the system, the DEP electrode closes the microfluidic chamber from the top and the electrode is in electrical contact with a liquid consisting of CTCs. To seal the microfluidic chip, the cover plate is placed from the top and M3 bolts are used to tighten the cover plate and top/bottom plates. To seal the system even more, an elastomeric o-ring (18.0 × 1.0 mm) is applied between the top plate and the cover plate. 

To apply an AC electric field, the system used two metal electrodes: The first one is called a counter-electrode that is placed at the top to the chamber and is micromachined into the top plate. This electrode is flat cylinder comprising aluminum with a diameter of 15 mm and thickness of 4 mm. At the top surface, it was drilled and a metal bolt was placed as a convenient method of connecting it with the source of the electric field. The second electrode is a commercially available sharp needle with a diameter of 260 μm, and it is made of stainless steel (Maanshan Bond Medical Instruments Co., Ltd., Maanshan, China). This electrode is placed in a specially milled channel (diameter 250 μm, slightly smaller than the diameter of the needle to ensure a close fit) at the very center of the SERS cavity (see Figure 1). This position and sharp point provide the electric field gradient in the area of the SERS platform.

The microfluidic chip has been designed using MasterCAM software and then micromachined with a computer numerical controlled (CNC) milling machine (ErgWind, type MFG4025P) using a 5 mm polycarbonate (PC) slab (Bayer) (see Appendix A for CAM visualization). Polycarbonate was chosen as the perfect material due to its transparency, high mechanical parameters (Young’s module), ease in machining, and the working and tested procedure of merging two plates of PC was feasible [61,62]. To join the milled PC slabs, we cleaned them in a baker filled with isopropanol, which was placed in an ultrasonic bath. Cleaning was conducted at 50 °C for 30 min. After drying with compressed air, the plates were pressed together at high temperatures (T = 130 °C) for 30 min with the use of a pneumatic press. Blunt-ended needles with an outer diameter of 0.8 mm were installed in the plane of the micromachined channels and additionally sealed with Epoxy glue. 

Standard polyethylene (LDPE) tubings (Micro Medical Tubing, Scientific Commodities, Inc., Lake Havasu City, AZ, USA) with an inner diameter of 0.8 mm were used for the interconnection of the chip with the syringe pump.

### 2.3. Setup for DEP Deposition of CTCs

An experimental setup for the dielectrophoretic deposition of CTCs consists of a custom-made microfluidic chip, a function generator combined with digital oscilloscope and a syringe pump. The schematic view of the setup is demonstrated in Figure 3A, while the picture of the setup is demonstrated in Figure 3B.

The source of the electric field was a function generator (SIGLENT, model SDG2042X) with two separate output channels: maximal frequency *f*_MAX_ = 40 MHz and maximal *U* = 20 V_pp_. The output signal was monitored in real time with a digital oscilloscope (type UNI-T, model UPO2074CS) connected to the output electrodes of the generator. The function generator was set in the high-impedance mode (HiZ) in order to work with the maximal frequency and maximal voltage amplitude, when the generator load is a two-electrode system for dielectrophoresis. The AC signal was delivered to DEP electrodes with shielded cables terminated with alligator clips. The DEP electrodes, as previously described, are placed directly above (DEP counter-electrode) and below (DEP electrode) the SERS active structure (see Figure 1c). The counter-electrode has a size of 15.0 mm in diameter and 4.1 mm in thickness. On the contrary, the bottom DEP electrode is 260 μm in diameter and is sharply ended (the diameter of the tip is 5 μm, see Figure 4).

The aspect ratio of the diameter of the DEP counter-electrode (15.0 mm) and electrode (5.0 μm) is 3000; therefore, there is a strong electric field gradient across the microfluidic chamber and SERS platform. To fill in the microfluidic chamber with liquid-containing CTCs, we used a syringe pump (NE-1000, New Era Pump Systems Inc., Farmingdale, NY, USA). The syringe was connected with the microfluidic chip by PE tubing, and to ensure a good and secure connection between the tube and the chip, we used a blunt needle at the chip’s inlet. The needle was placed between the top and bottom plate in the axis of the microfluidic channel so the CTCs cells do not stop or sediment at the constrictions of the tubing/microchannels. Because of the thin, lower DEP electrode, we placed the entire microfluidic chip in a 6-arm soldering station, which made access to both electrodes easier (see Figure 3B).

### 2.4. SERS Platforms and Measurements

The SERS-active platforms are based on silicon subjected to laser ablation. The procedure was described in detail by Szymborski et al. [63] The initial step involves cutting the silicon wafer (Łukasiewicz–the Institute of Microelectronics and Photonics, Warsaw, Poland) at the thickness of 525 ± 25 μm into small, typically 3.0 mm × 3.0 mm, squares by using a mechanical saw. Next, the modification of the silicon surface is performed by using a femtosecond laser working at a wavelength of λ = 1030 nm, whereas the repetition rate is 300 kHz and pulse width is 300 femtoseconds. These are the optimum parameters for the surface treatment of silicon by a femtosecond laser that provide a surface roughness suitable for the enhancement of the Raman signal. Finally, to complete the preparation of the SERS substrate, 100 nm layer of silver was sputtered using the PVD device (Quorum, Q150T ES, Laughton, UK). The SERS platforms are placed in sterile Petri dishes to avoid contamination from the air.

SERS measurements were performed using Wasatch (WP 830) Raman spectrometer. It is a fully modular, benchtop-type spectrometer equipped with separate 830 nm, multimode, power-tunable laser with 500 mW of maximum output power. The laser module, integrated with a power supply within the same enclosure, is connected to an external probe with an optical fiber. Spectrum collection is performed with a probe that consists of the main housing, equipped with a 40.0 mm tube spacer. This, in turn, is equipped with a Plano-Convex lens with a diameter of 9.00 mm that has an Effective Focal Length (EFL) of 18.00 mm (at 587.6 nm excitation wave). The use of this probe allows obtaining a spot size laser on a sample with a diameter of about 170 μm, and the backscattered radiation collected by it passes through a series of laser band pass filters, long pass/edge filters, optical fiber and after another series of long pass/edge filters falls onto the 25 μm slot and then to the detector. A spectral image is obtained in the range 250–1850 cm^−1^ with a spectral resolution below 8 cm^−1^. The optimal working temperature of the detector is between −15 °C and +15 °C and is controlled and maintained with the thermoelectric effect. 

To avoid a release of ions from the surface of CTCs to 0.1 PBS, the suspension of the cells in the buffer was prepared for a maximum of 2 min before the start of the experiment. SERS measurements were recorded repeatedly to obtain 15 single measurements for each sample. 

### 2.5. Numerical Calculations

#### 2.5.1. Theory of DEP

In dielectrophoresis (DEP), a non-uniform electric field induces a dipole moment on the particle due to the electrical polarization of the particle with the surrounding medium. As a result of this polarization; the DEP force (FDEP) acts on the particle relative to the areas of high intensity (positive DEP or pDEP) or low intensity (negative DEP or nDEP) of the electric field gradient. The dielectrophoretic force for the spherical particles is defined as follows [64]:$$F_{DEP}=2\pi r^{3}\varepsilon_{m}Re\left[f_{CM}\right]\nabla {\left|E_{rms}\right|}^{2}$$
where *r* is the radius of the particle, *ε_m_* is the permittivity of the medium, *Re*[*f_CM_*] is the real part of the Clausius–Mossotti (*CM*) factor, and ∇|*E_rms_*|^2^ is the magnitude of the electric field gradient (which depends on the geometry of the electrodes and the applied voltage). The effective polarizability of the particles with respect to the suspending medium determined by the *CM* factor is defined as follows [64]:$$f_{CM}=\frac{\varepsilon_{p\;}^{*}-\varepsilon_{m}^{*}}{\varepsilon_{p}^{*}+2\varepsilon_{m}^{*}},$$
where $\varepsilon_{p}^{*}$ and $\varepsilon_{m}^{*}$ are the complex permittivity (*ε** = *ε* − i(σ/ω)) of the particle (*p*) and the surrounding medium (*m*), and i = $\sqrt{-1}$. By changing the frequency (ω) of the applied electric field and the electric conductivity (σ), the effects of complex permittivity can be controlled in addition to the direction in which the particle moves (pDEP or nDEP effect). The point at which the DEP force switches is called the crossover frequency (*f*_c_) and their value is crucial for precisely knowing the pDEP and nDEP range of frequencies (see Figure 5 with marked pDEP and nDEP regions). The crossover frequencies can be easily calculated with a tailor-made software, e.g., myDEP, with an implemented Clausius–Mossotti equation.

#### 2.5.2. Crossover Frequencies (*f*_c_) and Distribution of Electric Field

To calculate the real part of the Clausius–Mossotti (Re[*f*_CM_]) factor for different parameters and models of cancer cells, we used the software developed by Cottet et al. [65]. myDEP is a desktop software written in Java, and it allows the study a dielectric response of particles (in our case, CTCs) applied to AC electric fields. The software consists of a Graphical User Interface (GUI) and a literature-based database of the parameters of cancer cells. 

myDEP was used to calculate the cross-over frequencies (*f*_c_) for MCF-7 and MDA-MB-231 cancer cells, thus assessing the regions, where the positive dielectrophoretic (pDEP) force will be acting on the CTCs in the direction of the SERS platform and depositing them on the center of the platform. We have used built-in data of both MCF-7 and MDA-MB-231 (see Appendix A) and an electrical conductivity of 0.1 PBS (0.19 S/m).

The distribution of the electric field inside chamber is essential, as the deposition of CTCs occurs only for the pDEP region. Therefore, the precise design of the shape and positioning of the electrodes is crucial in terms of optimal depositions of the CTCs on the platform. For this reason, a 2D numerical model of the dielectrophoretic chamber and SERS platform was developed in COMSOL Multiphysics 5.4a. COMSOL is a general-purpose simulation software and uses the Finite Element Method (FEM) to calculate the potential, the distribution of electric field and visualizes the gradient of the electric field inside the dielectrophoretic chamber. The standard AC/DC module and a stationary model was used for numerical calculations. In solving any problem using COMSOL, there are three main stages:The pre-processing stage is where the fundamentals of the model are developed (e.g., geometrical structure) and parameters are determined (e.g., type of materials, domain and boundary conditions and meshing of the geometry). The geometrical dimensions for the calculations are taken from CAD/CAD designs or from SEM observations in the case of the steel needle (Figure 4).The solution stage uses mathematical equations (e.g., Partial Differential Equations) and information on the type of problem (stationary or time-dependent) to solve the problem.The post-processing stage is where the results are shown and the user can generate 1D, 2D or 3D plots.

### 2.6. Microscopic Characterization

Scanning electron microscopy (SEM) was used to visualize the shape and curvature of the stainless-steel needle used as the DEP electrode and to record CTCs on the surface of the SERS platform after dielectrophoretic depositions. The observations were performed under high vacuum using the FEI Nova NanoSEM 450 (Hillsboro, OR, USA). The accelerating voltage was from 2 kV to 10 kV. The samples were attached to SEM stabs with carbon tape or silver paint and observed without any additional layers of gold or carbon on the surface.

The photographs were taken with a digital single-lens reflex (DSLR) camera (80D, Canon, Japan) equipped with Canon 40 mm/2.8, and Tamron 90 mm/2.8 Macro lenses. RAW files were developed with ON1 Photo RAW software (ON1, Portland, OR, USA).

## 3. Results

### 3.1. Numerical Calculation for the DEP Deposition of the CTCs

#### 3.1.1. Crossover frequencies (*f*_c_) and pDEP Region

A crossover frequency (*f*_c_) can be defined as the frequency where the Clausius–Mossotti factor (and the strength of dielectrophoretic force) changes from negative (positive) dielectrophoresis to zero and then becomes positive (negative) dielectrophoresis. Thus, exactly for *f*_c_, the dielectrophoretic force is zero and the object (in our case, CTC) does not move towards or away from a strong electric-field gradient. The precise knowledge of the values of the crossover frequencies described in the Section 2.5.1 is crucial in terms of the efficient deposition of the CTCs on the surface of the SERS platform. The deposition works in the designed system only for pDEP, i.e., the situation where the CTCs are working dielectrophoretic forces directed into the high-intensity gradient of the electric field. The real part of the Clausius–Mossotti factor was calculated in myDEP software, and the plot for the MCF-7 and MDA-MB-231 is presented in Figure 5, whereas the crossover frequencies are summarized in Table 1.

The plots of the Re(CM) for both CTCs are similar. Frequencies lower than *f*_c1_ on the CTCs involve acting negative dielectrophoretic forces; thus, the CTCs are repelled from the high-intensity electric field (negative dielectrophoresis, nDEP). The *f*_c_ for both CTCs are close to 200 kHz (see Table 1 for exact numerical value), and above that frequency, the pDEP should be visible. The next crossover frequencies, which mean the change of pDEP to nDEP, are 248 MHz for MCF-7 and 482 MHz for MDA-MB-231. Such large values are beneficial from an experimental point of view: we have a large range of frequencies within which we can study its effect on the deposition of CTCs. The limitation here is, of course, the range of the signal generator, which, for the maximum voltage range, can only be used up to 20 MHz. Therefore, in both cases, the effect of pDEP will be studied for frequencies between 200 kHz and 20 MHz.

The highest value of dielectrophoretic strength (for both CTCs) is to be expected at the plateau of the pDEP range (between 5 MHz and 20 MHz (the maximum useful value obtained from the generator)).

#### 3.1.2. FEM Analysis of Electric Field Distribution

To model the potential and electric field, we have chosen 2D models instead of 3D models. The dielectrophoretic chamber, electrodes and SERS platform are symmetrical; therefore, the model can be simplified to 2D and the accuracy can still be at an acceptable level. The geometry was created using the COMSOL drawing tool and all dimensions were maintained in accordance with the original design. The figure demonstrating the materials used in COMSOL, their dielectric constant (*ε*_*r*_) and the mesh for calculations is provided in the ESI (see Appendix A). The boundary conditions were set to the steel needle (+10 V) and to the counter-electrode (−10 V). The electric potential is demonstrated in Figure 6a.

The plot shows that the tip of the DEP electrode (sharp needle) and the width was set to 3 mm, e.g., the width of the SERS platform. The potential has the highest value at the electrode and decreases relative to the counter-electrode. The magnification of the tip of the electrode is demonstrated in ESI (see Appendix A). Using the potential, we can calculate the normalized electric field distribution between the two electrodes. Figure 6b shows the magnification of the DEP electrode with the visible gradient of the electric field (with the highest value at the top of the needle). The EF distribution under the entire SERS platform is demonstrated in ESI; see Appendix A.

### 3.2. DEP-Based SERS System for Detection of CTCs

#### 3.2.1. DEP Parameters and Its Influence on SERS Detection

To establish the best conditions for CTC’s trapping and the excellent performance of the DEP-SERS device, the DEP captured process parameters such as frequency of the alternating electric field (*f*), the electrical conductivity of the medium (σ) in which the cancer cells were suspended, and the time of deposition (*t*) that has been investigated.

We examined the performance of our DEP–SERS system for the molecular recognition by measuring the SERS responses of MCF-7 and MDA-MD-231 breast cancer cells in a 0.1 PBS buffer. All SERS spectra (see Figure 7) exhibit vibrations characteristic for cancer cells [66,67] but with distinctly varying intensities depending on the parameters of the DEP deposition. The strongest SERS band at 650 cm^−1^ was selected for comparing the efficiency of MCF-7 and MDA-MD-231 trapping at different DEP-capturing conditions. Figure 7 demonstrates the influence of the deposition time and applied frequencies to SERS responses of both types of studied cancer cells. As observed in Figure 7A,B, the optimal time of deposition for MCF-7 is 20 min, whereas for MDA-MB-231, it is 7 min (Figure 7C). In Figure 7D, we present the intensity of the band 650 cm^−1^ as a function of applied frequency (*U* = 20 Vpp) for both MCF-7 and MDA-MB-231. The regions of nDEP and pDEP are marked. The use of the frequencies from the pDEP region results in a higher intensity of the selected band (which is representative for the whole SERS spectra). 

Appendix A compares the SERS response within the nDEP and pDEP regions of frequency. As observed, the SERS spectra were recorded below the crossover frequency (*f*_c_) of the MCF-7, e.g., 197 kHz. The used frequency, i.e., 15 kHz, was observed in the nDEP region (yet high enough to avoid electrophoretic and polarization effects). In comparison, 20 MHz is an optimal frequency in the pDEP region, which is evident when we compare the intensities of the band at 650 cm^−1^ (3800 cps to 288 cps). A similar effect was also present for MDA-MB-231. 

In the next step, we compared the efficiency of SERS detection of MCF-7 and MDA-MD-231 cancer cells with pDEP (for optimal established parameters of frequency, voltage and time) and without DEP deposition. Figure 8A,B present the selected SERS responses for (a) the spontaneous adsorption of studied cancer cells onto the SERS substrate and (b) the established capturing conditions in the pDEP-SERS system. 

When the electric field is applied, the pDEP force causes the deposition of cancer cells on an SERS platform. The SERS spectra of cancer cells are determined by the molecular species that are in proximity to the nanoplasmonic substrate and reveal common spectroscopic features characteristic to the constituent of the eukaryotic cell [68]: nucleic acid, proteins, and lipids. 

The recorded SERS spectrum of MCF-7 presented in Figure 8A is dominated by the bands at 650, 720, 790, 880, 910, 1005, 1030, 1120, 1330, 1445, and 1665 cm^−1^. The intensive band at 656 cm^−1^ corresponds to the C-C vibrations of tyrosine. Aromatic amino acid contributions appeared at around 1002 cm^−1^ (phenylalanine). The vibrational modes of nucleic acid are present at 789 and 1090 cm^−1^. The shoulders around 1256 cm^−1^ and very intensive band at 1665 cm^−1^ are assigned to amide I bands. As observed, all cells also reveal their own individual spectral signatures; e.g., the band at 910 cm^−1^ corresponded to the C-C vibration of proline, and valine [65] can be seen in MCF-7 cells but not in MDA-MD-231. The relative intensities of some bands can also serve as the method of cell differentiation. For example, very strong bands at 1445 and 1665 cm^−1^ in the spectrum of MCF-7 are very weak in the SERS spectrum of MDA-MD-231 cells. These dissimilarities enable the identification of circulating tumor cells. For more details and precise band assignments, please refer to Table 2.

Additionally, SEM images were registered to monitor the presence of cancer cells on the SERS platforms. Figure 8C,D demonstrate the SEM images of the SERS platform with MCF-7 and MDA-MD-231 cancer cells, respectively. CTCs were deposited on the platforms with optimal parameters (MCF-7: *f* = 20 MHz, *U* = 20 Vpp, *t* = 20 min, whereas MDA-MD-321: *f* = 5 MHz, *U* = 20 Vpp, *t* = 7 min).

As a comparison, we have placed the suspension of the CTCs in 0.1 PBS solution on the SERS platform and deposited them on SERS surfaces with an evaporation rate of the 0.1 PBS at room temperature. The spectral intensities and resolution of the collected SERS spectra (Figure 8A(a),B(a)) are significantly lower (for example, the intensity of the marker band for MCF-7 at 650 cm^−1^ has 290 counts per second (cps), while in the case of dielectrophoretic force-induced trapping, it reaches up to 3800 cps). Both cancer cells and cases are summarized in the Table 3.

#### 3.2.2. The SERS-DEP Detection Sensitivity

For the sensitivity study, various concentrations of MDA-MB-231 cancer cells (8, 15, 20, 40, 65, 100, 300, 500, 840 and 1400 cancer cell in 1 mL of 0.1 PBS) were examined by the SERS-DEP protocol.

As shown in Figure 9A, intensity of the SERS spectra increased as the concentration of MDA-MB- 231 cells increased. Figure 9B shows that the SERS intensity of the marker band at 650 cm^−1^ increases linearly with MDA-MB-231 cell concentrations between 8 and 500 cells/mL (R^2^ = 0.946). The limit of detection (LOD) is 20 cells/mL, which reflects the clinical potential and usefulness of the developed approach [4]. 

To summarize, the obtained data clearly indicate that the pDEP-SERS approach can be used as a novel method for capturing and identifying CTCs. As the amount of single CTCs in peripheral blood is rare, the highly efficient cell enrichment and single-cell capture methods are essential for screening target cells. To evidence the usefulness of our approach for deposing CTCs from blood plasma, we have performed experiments, and the results are presented in Figure 10. 

Figure 10A reveals the examples of SERS spectra of the MCF-7 cells captured and detected from blood plasma samples using the pDEP-SERS device while using parameters calculated for blood in pDEP configuration. The intensity of the SERS spectrum of MCF-7 cancer cells trapped in blood is not such higher than in the case of a buffer solution (see Figure 7A); however, the spectral fingerprints characteristic of MCF-7 are present in the spectrum: the bands at 650 cm^−1^ (C-C twisting mode of tyrosine), 1417 cm^−1^ (C=C stretching in quinoline ring) and 1452 cm^−1^ (structural protein modes of tumors). It should be noted that, in the SERS spectrum under pDEP forces, bands corresponding to blood plasma (804, 1224 and 1303 cm^−1^) also appeared, but they were evidently less intensive in comparison to the SERS spectrum of MCF-7 cancer cells or these bands may be also overlapped by the strong bands of trapped and concentrated cancer cells. Figure 10B presents the selected SERS responses during the pDEP deposition of MDA-MB-231 cancer cells in blood plasma after the application of DEP with different frequencies (from 0.1 MHz to 20 MHz). As one can see, for the frequency equal to 20 MHz, the fingerprint of MDA-MB-231 is clearly detectable in the collected SERS spectrum (bands at: 650, 810, 1450 and 1650 cm^−1^). In contrast, with the electric field applied to the blood plasma in the DEP-SERS device, SERS signatures are mainly measurable.

Our preliminary results indicate that the proposed DEP-SERS device introduces a new possibility for CTCs. In order to better understand the dependence of the frequency on the DEP process in whole blood and blood plasma, more advanced experiments will be made in future.

## 4. Conclusions

In this article, we demonstrate the implementation of positive dielectrophoresis (pDEP) relative to SERS-based depositions and the detection of breast cancer cells (MCF-7 and MDA-MB-231) on a solid-state SERS platform. The dielectrophoretic effect allows for a controlled deposition of CTCs on the SERS platform (which is localized directly above the DEP electrode), whereas the SERS technique enables very sensitive vibrational-based cancer cells identifications. 

We have numerically calculated (myDEP, COMSOL Multiphysics) and then experimentally optimized the parameters of the pDEP process, such as the frequency of the alternating electric fields (*f*), applied voltage (*U*) and the time of deposition (*t*) of the CTCs in the 0.1 PBS buffer. The optimal parameters for MCF-7 were *f* = 20 MHz, *U* = 20 Vpp and *t* = 20 min, whereas for MDA-MD-231, the optimal parameters were *f* = 5 MHz, *U* = 20 Vpp and *t* = 7 min. The use of optimal parameters for pDEP results in an increase in spectral resolutions and intensities of the selected marker band up to 13 times. The results were obtained for a 0.1 PBS buffer, and we also provided a proof-of-concept for the deposition and identification of CTCs in the blood plasma. The calculated limit of detection (LOD) for MDA-MB-231 in 0.1 PBS buffer solution was 20 cells/mL.

The proposed combination of pDEP and solid-state SERS platform in a microfluidic device will be further developed for whole blood samples with CTCs towards the detection of cancer cells in real clinical samples.

## Figures and Tables

**Figure 1 biosensors-12-00681-f001:**
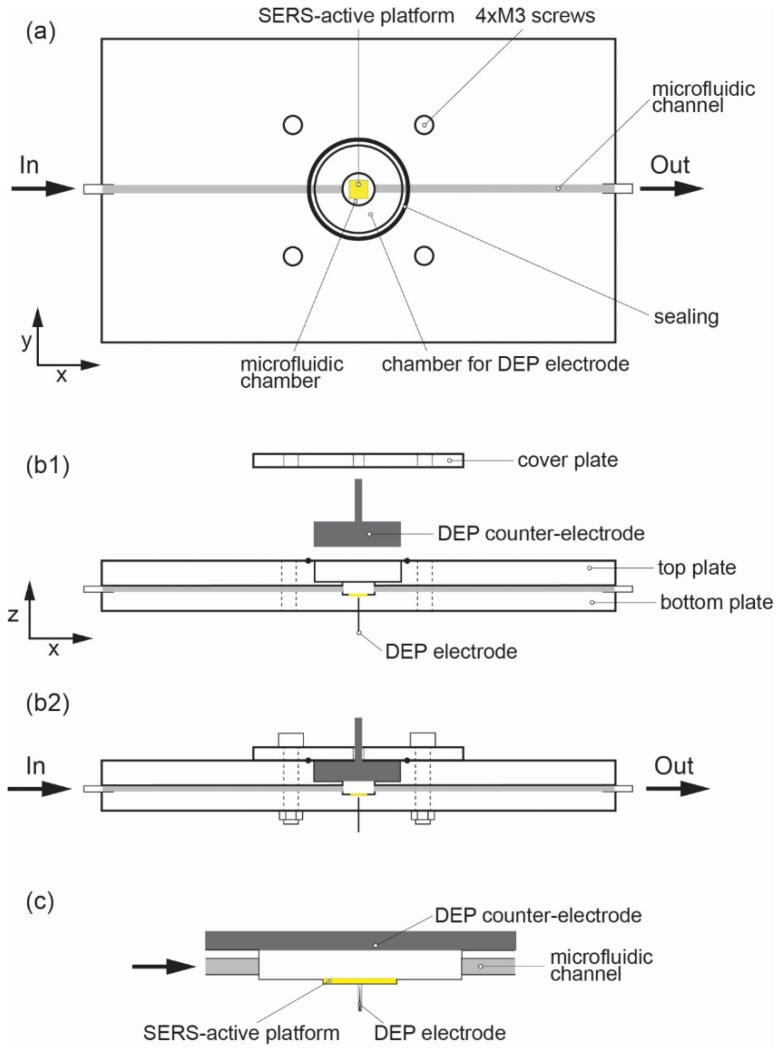
Schematic view of the microfluidic chip for the dielectrophoretic deposition of CTCs on the surface of the SERS platform (**a**), cross-section of the chip before assembly (**b1**) and after placing cover plate on the top plate (**b2**), detailed view of the microfluidic chamber with the DEP electrode under the SERS platform and counter-electrode at the top of the chamber (**c**).

**Figure 2 biosensors-12-00681-f002:**
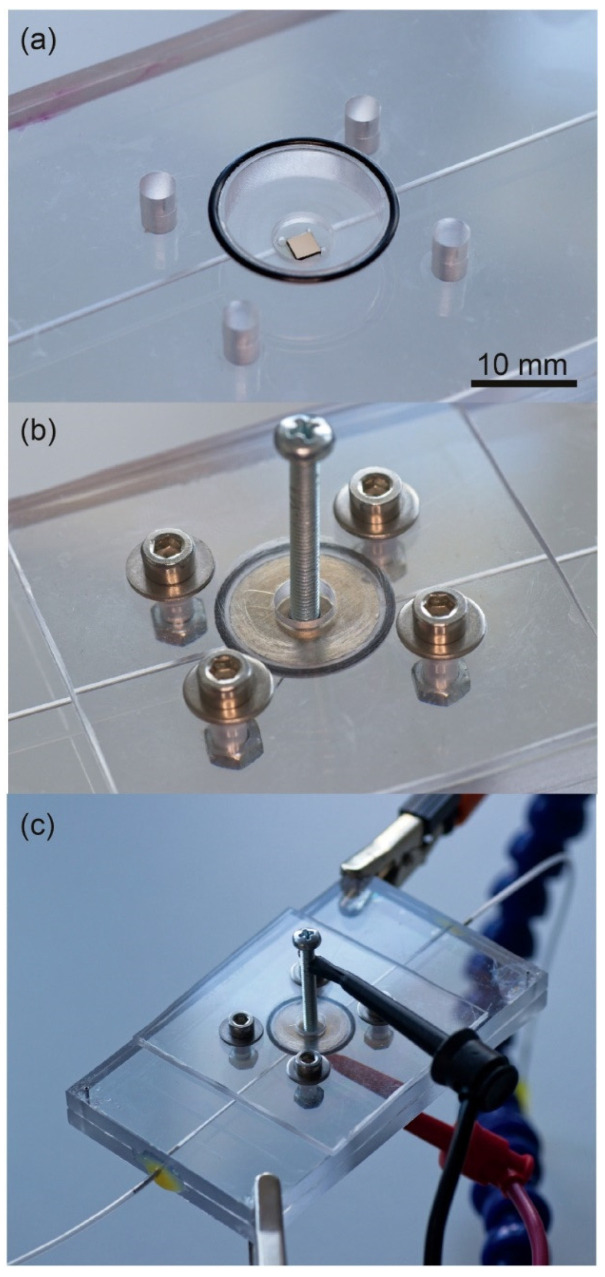
Bottom plate of the microfluidic chip for the dielectrophoretic deposition of CTCs with the SERS platform at the very center and chamber for the DEP counter-electrode and sealing (**a**); bottom plate with the DEP electrode closed with a top plate and assembled with four M3 bolts (**b**); view of the assembled microfluidic chip with a connected signal generator and PE tubing running to a syringe pump (**c**).

**Figure 3 biosensors-12-00681-f003:**
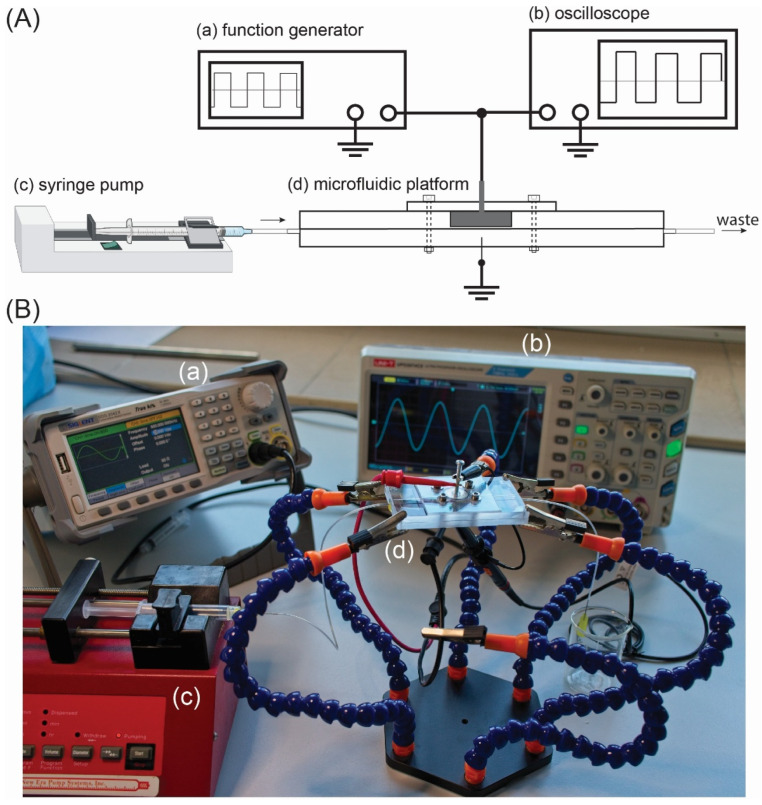
(**A**) The experimental setup for deposition of CTCs on SERS platform: the AC electric filed is generated by a function generator (a), monitored with a digital oscilloscope (b) and applied to DEP electrode and counter-electrode, which are a part of the microfluidic platform (d). CTCs can be introduced into microfluidic platform via the syringe pump and PE tubing (c). The microfluidic platform in not at scale. (**B**) The experimental setup used for dielectrophoretic deposition of CTCs on the SERS platform. The DEP chip in placed on a 6-arm PCB holder. The DEP electrodes are connected to a signal generator (a) and digital oscilloscope (b). CTCs are introduced to microfluidic chip (d) via the syringe pump (c).

**Figure 4 biosensors-12-00681-f004:**
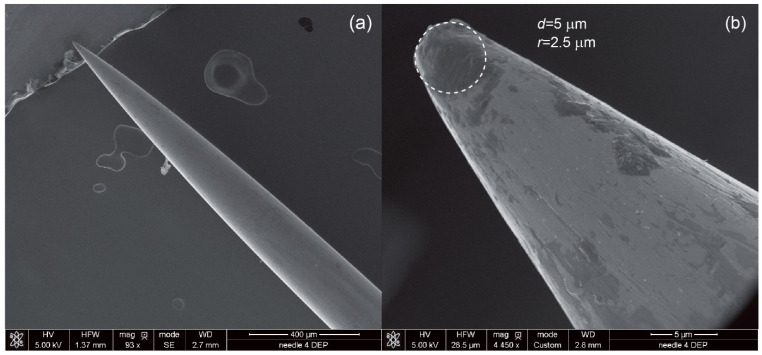
DEP electrode placed from the bottom of the SERS platform. The small diameter (5 μm in comparison to the diameter of the needle at 260 μm) results in a high electric-field gradient, which is required for effective dielectrophoretic deposition of the CTCs. (**a**) the general SEM view of the needle, (**b**) higher magnification with marked diameter of the end of the needle.

**Figure 5 biosensors-12-00681-f005:**
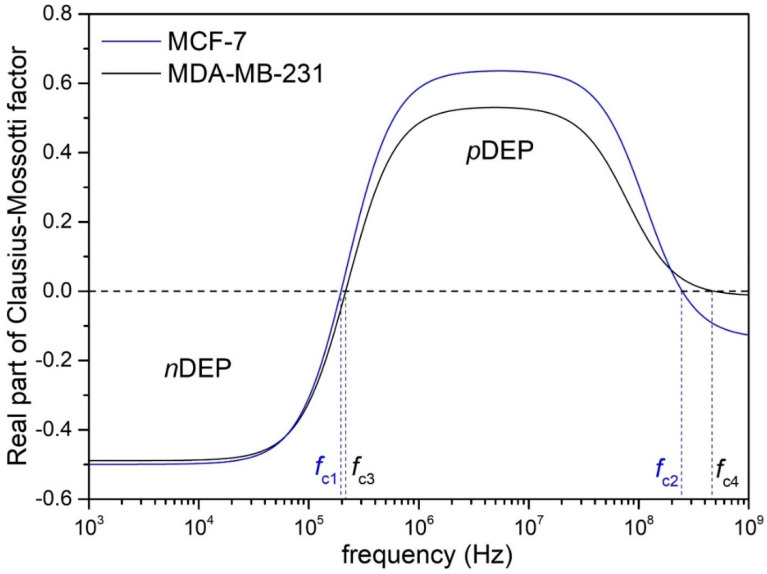
The real part of the Clausius-Mossotti factor (Re(CM)) as a function of the frequency of alternative electric field (AC EF). The figure consists of two CTCs: MCF−7 and MDA−MB−231. Negative dielectrophoresis (nDEP) and positive dielectrophoresis (pDEP) are indicated in the figure. The calculations were performed for 0.1 PBS using myDEP software.

**Figure 6 biosensors-12-00681-f006:**
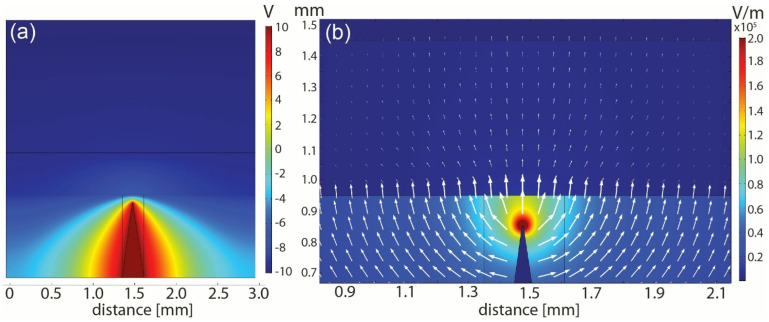
FEM analysis of the dielectrophoretic deposition system in the very center of the microfluidic chip. (**a**) The potential (U) between the DEP electrode (needle) and the counter−electrode (very top of the figure). (**b**) The electric field (EF) distribution between the two electrodes.

**Figure 7 biosensors-12-00681-f007:**
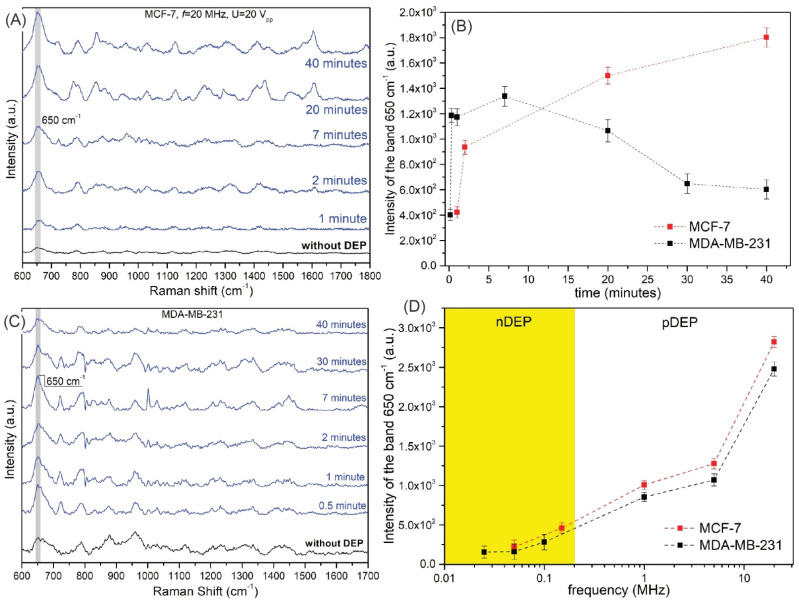
(**A**) The SERS spectra of the MCF−7 cancer cells in the function of the deposition time via pDEP force for *f* = 20 MHz and *U* = 20 Vpp. (**B**) The averaged intensity of the band at 650 cm^−1^ of the MCF−7 (*f* = 20 MHz, *U* = 20 Vpp) and MDA−MD−231 (*f* = 5 MHz, *U* = 20 Vpp) for different deposition times (0.5 min to 40 min). (**C**) The selected SERS spectra of the MDA−MB−231 cancer cells in the function of the deposition time via pDEP force for *f* = 20 MHz and *U* = 20 Vpp. (**D**) The averaged intensity of the band at 650 cm^−1^ of the MCF−7 (*U* = 20 Vpp, *t* = 20 min) and MDA−MD−231 (*U* = 20 Vpp, *t* = 7 min) for different applied frequencies. All measurements were performed in the 0.1 PBS solution.

**Figure 8 biosensors-12-00681-f008:**
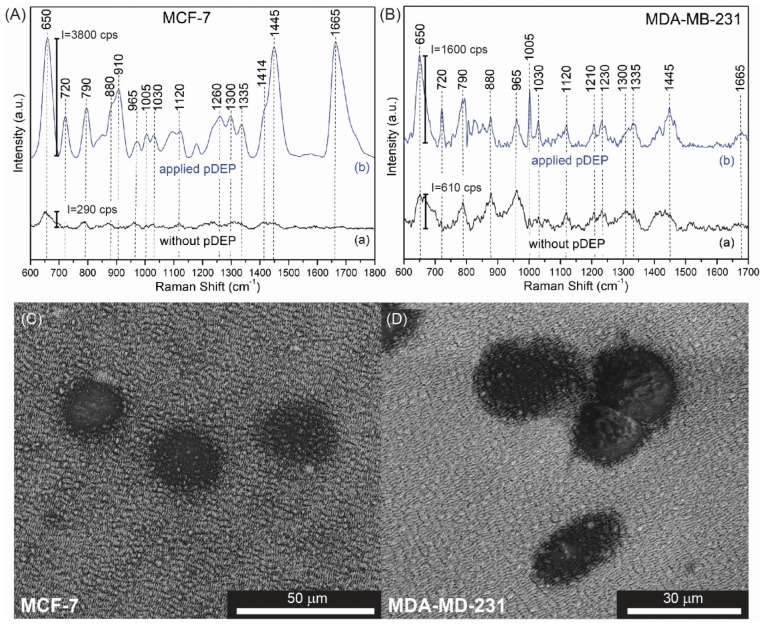
The SERS spectra of (**A**) MCF−7 and (**B**) MDA−MD−231 recorded in the pDEP−SERS device without (a) and with (b) the application of dielectrophoretic trapping of CTCs. The corresponding SEM images of (**C**) the MCF−7 and (**D**) MDA−MD−231 deposited on the surface of the SERS platform with this dielectrophoretic effect.

**Figure 9 biosensors-12-00681-f009:**
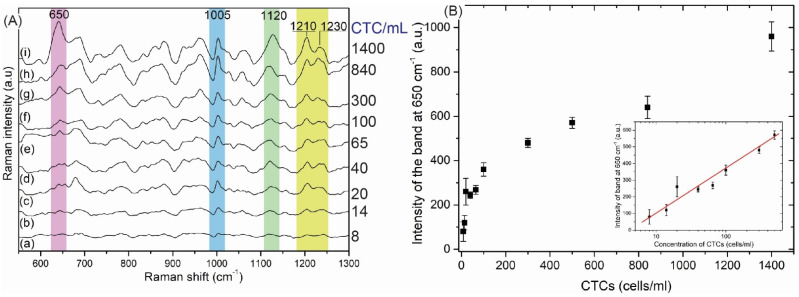
(**A**) SERS signals of selected concentrations of MDA−MB−231 ranging from 8 to 1400 cells/mL. (**B**) The relationship between the intensity of the marker band at 650 cm^−1^ versus the concentration of MDA−MB−231 in the entire analyzed range of CTC concentrations with an insert showing the linearity of SERS responses between 8 and 500 cells/mL.

**Figure 10 biosensors-12-00681-f010:**
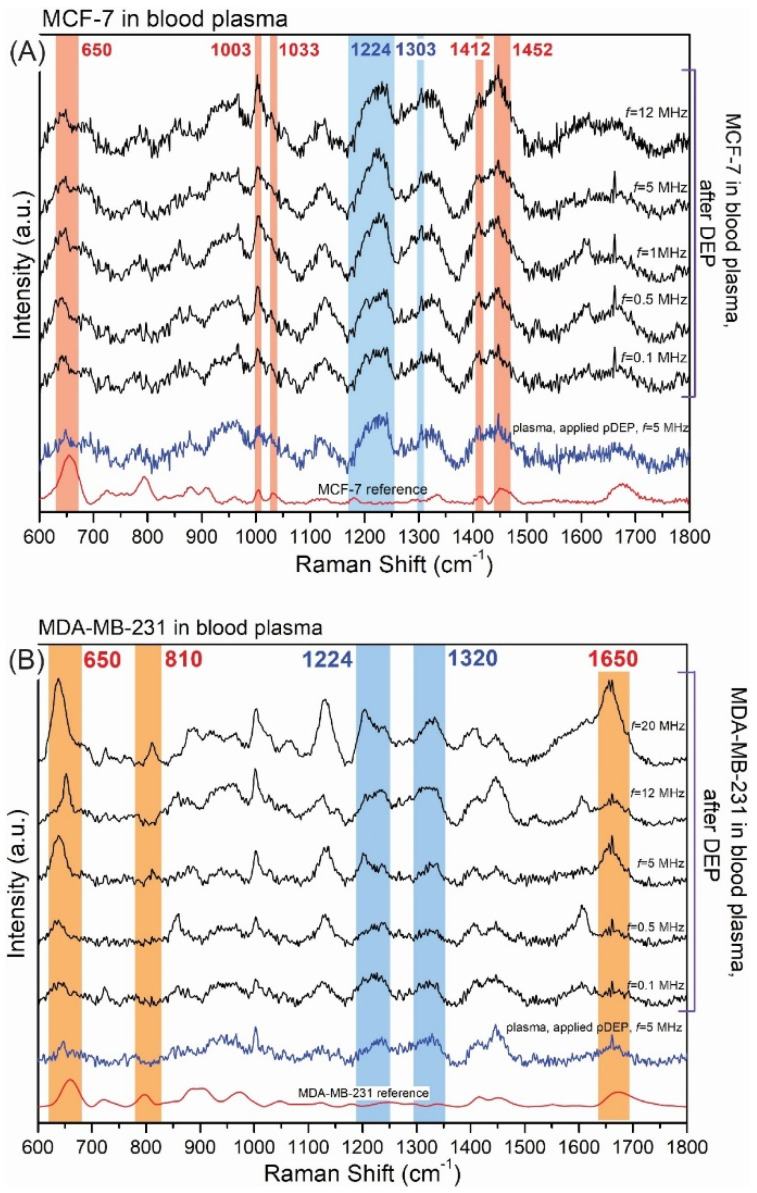
The SERS spectra of the reference cancer cells, blood plasma after applied DEP and cancer cells in the blood plasma after the application of DEP with different frequencies (from 0.1 MHz to 20 MHz) recorded for (**A**) MCF−7 and (**B**) MDA−MB−231.

**Table 1 biosensors-12-00681-t001:** Crossover frequencies (*f*_c_) of MCF-7 and MDA-MB-231. The calculations were performed for 0.1 PBS with an electrical conductivity of σ_m_ = 0.19 S/m.

Type of CTCs	*f*_c1_ (kHz)	*f*_c2_ (MHz)
MCF-7	196.83	248.64
MDA-MB-231	216.82	482.33

**Table 2 biosensors-12-00681-t002:** Assignment of SERS bands depicted in Figure 8 [69,70,71,72].

Observed SERS Band (cm^−1^)	Protein	Lipids	Nucleic Acid	MCF-7	MDA-MB-231
650	Tyr (C-C twist)			**+++**	**+++**
720–730	Trp	C-N head group choline (H_3_C)_3_N+	A	**+++**	**+++**
790			PO_2_ symm	**+++**	**+++**
827	Structural protein modes of tumors			**+**	**++**
850–880	Tyr, Pro			**+++**	**++**
910	C-C str alpha-helix, Pro, Val			**+++**	**+**
960	CH_3_ def	CH_3_ def		**+**	**++**
1005	Phe			**++**	**+++**
1030	Phe	CH_2_CH_3_ bending modes of lipids		**++**	**++**
1090	C-N stretch	CC str chain, C-O str	PO_2_ symm	**++**	**+**
1120	C-N str bk	porphyrin		**++**	**++**
1210	C-C_6_H_5_ str in phenylalanine tyrosine and Amide III (beta sheet)			**+**	**++**
1230–1240	Amide III			**++**	**++**
1265–1270	Amide III (random coil)	=CH def			
1300–1308	CH_3_ def, collagen	CH_3_CH_2_ twist	G	**++**	**++**
1335			A, G	**++**	**++**
1414–1418	C=C stretching in quinoid ring			**++**	**++**
1445–1450	structural protein modes of tumors			**+++**	**+++**
1552			A, G	**+**	**-**
1585–1600	Phe, Tyr			**+**	**+**
1650–1665	Amide I	C=C str		**+++**	**++**

**+** weak intensity, **++** medium intensity, **+++** high intensity.

**Table 3 biosensors-12-00681-t003:** Summary of efficiency of deposition of the CTCs (applied pDEP and spontaneous, gravitational deposition—no DEP) calculated for intensity of marker band at 650 cm^−1^.

Cancer Cell	Intensity of Band at 650 cm^−1^ (cps)		Aspect Ratio (I_pDEP_/I_noDEP_)
	*p*DEP	no DEP	
MCF-7	3800	290	13.1
MDA-MB-231	1600	610	2.6

## Data Availability

Not applicable.

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
