# Peer review of "Dielectrophoresis-Based SERS Sensors for the Detection of Cancer Cells in Microfluidic Chips"

_biosensors, 2022, doi:10.3390/bios12090681_

Round 1

Reviewer 1 Report

The work is well performed and fairly well presented. 

It is unclear what is the relationship between the figure 7C and 8A spectra. If they are supposed to represent the signal from the same object (MCF-7), why are they different?

I was unable to find the electonic supplementary figures the authors refer to in several places in the manuscript.

Author Response

Dear Reviewer,

Based on the Reviewers' comments for the manuscript and in accordance with their suggestions, we enclose a revised version of the manuscript entitled:

‘Dielectrophoresis-based SERS sensor for detection of cancer cells in microfluidic chip’

Tomasz R. Szymborski, Marta Czaplicka, Ariadna Nowicka, Joanna Trzcińska-Danielewicz, Agnieszka Girstun, and Agnieszka Kamińska

All changes in the main manuscript have been made in red. 

We thank you for your valuable comments.

Authors

Reviewer #3

The work is well performed and fairly well presented.

We thank the Reviewer for the nice words.

It is unclear what is the relationship between the figure 7C and 8A spectra. If they are supposed to represent the signal from the same object (MCF-7), why are they different?

We thank the Reviewer for this comment.

For a clear and reasonable presentation of the obtained results, we decided to include in Figure 7 the time relationships represented in the Sers spectra for both tested CTCs cell types. Figure S5 from ESI is now in place of Figure 7C. Fig. 7C (corrected) has been moved to ESI as Fig. S5. Indeed, there is a mistake in the Figure 7. We have exchanged SERS spectrum for 7 minutes (now is in the Figure S5), which is according Figure 7B not long enough to efficiently deposit MCF-7 on the surface of the SERS platform, to SERS spectrum recorded for 20 minutes. This time was selected as it gave the most intense and reproducible spectral fingerprint of MCF-7 cancer cells.

I was unable to find the electronic supplementary figures the authors refer to in several places in the manuscript.

We thank the Reviewer for this comment.

We submitted the ESI as a PDF file together with the manuscript (DOC file) – they were both zipped into a single archive. We understand that might cause the problem with downloading the file. To avoid that, this time we have created additional catalogue were the newest ESI file is downloadable. In case it will be not in the MDPI system, you may reach it here:

http://www.szymborski.net/FILES/Biosensors_DEP/

Reviewer 2 Report

A brief summary

This paper describes the careful development of a custom-developed microfluidic biosensor platform which integrates positive dielectrophoresis (pDEP) electrokinetics with surface-enhanced raman spectroscopy to detect circulating tumour cells (CTCs) which have been resuspended within an electrolytic buffer of known solute concentration. The pDEP effect is used to induce the motion of CTCs within the electrolytic suspension towards a surface. Upon contact and subsequent adherence to the material surface, surface-enhanced raman spectroscopic detection of the CTCs can be performed rapidly (in minutes). Effective theoretical and computational analysis was performed to identify the ideal frequency-dependent parameters required for the best performance of the dielectrophoretic arm of the platform.

General concept comments

·        Has the limit of detection of this system been characterized? As mentioned, CTCs are normally scarce in real samples, thus detection methods that may be of future clinical relevance should demonstrate the ability, or potential ability, to detect low numbers of CTCs. If no such characterization exists, performing some serial dilutions of the stock suspension to check the limit of detection is recommended.

·        Was the conductivity of your 0.1x PBS solution recorded? Details of this conductivity should be included for reference since any deviations from the expected 0.16 S/m conductivity would influence the ideal pDEP conditions for cell adherence to the SERS substrate.

·        Did you keep the cells in 0.1x PBS for a controlled time before starting the experiment? How long was this time? This is a useful detail for readers to know. Researchers who want to recreate this experiment may understand that the CTCs will experience osmotic pressure in a buffer of low conductivity (ie, a buffer that is not of physiological conductivity) such as 0.1x PBS. The effect is that over time, CTCs will release ions into the 0.1x PBS buffer and their conductivity will slowly change. Consequently, the ideal AC field conditions (voltage, frequency) required for pDEP-based deposition may change over time.

·        Built-in values of electrical parameters were used in myDEP software for the simulations. What were these values? Do they relate well to the values agreed on the literature? Please provide details of the parameters used (ie, in the text or in a table) for the cells (and buffer) and provide references which validate the used cell parameters.

·        There are references to supplementary figures and tables, but the data is not visible or downloadable. Please ensure that this file has been generated.

·        In Table 1, can the authors explicitly state which buffer was used and include in brackets its electrical conductivity? Without reading the text in detail, the reader cannot easily identify which buffer these cells will have said DEP crossover frequency.

·        In Figure 6, can the font size be increased for all axis labels. This is possible to edit in COMSOL. Axis titles should also be added.

·        There are some instances where the authors say PBS buffer was used, but presumably mean that 0.1x PBS was used. This is confusing: it leads the reader to believe that they stopped using 0.1x PBS and moved to 1x PBS. Nonetheless, the authors should confirm that all experiments were performed in 0.1x PBS.

·        The statement that the authors have demonstrated the deposition of CTCs from blood plasma is a significant statement. This figure should not be included in the ESI but instead in the main text.

·        In future, this should be integrated with another CTC isolation method. In it’s current format, another method of retrieving CTCs from real samples is required before this detection method can be used, given the dependence of the method on using an electrolyte buffer of a known conductivity. Can the authors comment of the ease of integration of this detection method with other CTC isolation/ separation methods?

Specific comments referring to line numbers, tables or figures that point out inaccuracies within the text or sentences that are unclear. These comments should also focus on the scientific content and not on spelling, formatting or English language problems, as these can be addressed at a later stage by our internal staff.

·        Lines 42-47: References 5 does not discuss all four of the mentioned CTC isolation/ separation methods, rather it only shows affinity-based separation of CTCs. Reference 6 does not seem a correct reference – it is more SERS focused. Please insert correct references covering all methods of CTC isolation.

·        Line 46: point (iv) begins with ‘Chapter’, which seems an incorrect word choice. I presume that this should read either ‘separation’ or ‘isolation’.

·        Lines 51-55: References 4,8 are not citations capable of validating all of the detection methods mentioned. Reference 4 is a review paper, not original work. Reference 8 discuss digital PCR-based detection of cfDNA and CTCs. Please identify appropriate original work for each of the different mentioned detection methods and include as references.

·        Lines 61-67: It may be better for readers if this paragraph is divided up into, eg, 4 sentences. It is currently made of two very long sentences making it more difficult to read.

·        Lines 68-69: The authors suggest that no “gold standard” method of CTC detection has been created yet. Can this statement be validated by adding references from other prominent sources in the field who agree with this synopsis?

·        Line 72: The authors suggest that DEP is the “perfect technique” for manipulation of cells/bacteria, but it is possible others in the field would contest this. It is perhaps better to suggest that this is “a technique” which has been used to manipulate cells/ bacteria, and then cite appropriate work.

·        Line 86: In the field of AC electrokinetics, the convention is to describe a cell/bacteria/or particle loaded solution as a mixture. This is based on Maxwell’s first publications describing his mixture theory. Using the terminology ‘varied solution’ to describe a mixture is not commonplace – it may be useful to change this to ‘mixture’ to avoid confusion and maintain use of the common language in this research area.

·        Line 92: The opening part of this sentence may be missing a word or two. Do you instead mean that “Since many cells have distinctive size and shape characteristics,…”?

·        Line 94: The word “a” should be added immediately before “DEP-based microfluidic device…”.

·        Line 106-107: Although it is well known that Microfluidics can perform benchtop laboratory processes like sample mixing, flow control, and separation in a miniaturized format, it is still necessary to provide references to original work which demonstrates such activities. Please add relevant citations.

·        Line 290: I assume that the author means to say “PBS x 0.1” rather than PBS x 10, which would be 100x more concentrated. There are also several references to the “PBS solution” which follow. The authors should instead ensure that these are defined as “0.1x PBS”.

·        Line 306: please define the abbreviation PDE, where talking about specific mathematical methods.

·        Lines 168, 216, 261: All references to “electric gradient” should be corrected to “electric field gradient”, because these two terms do not have the same meaning.

·        Line 273: The terms i and σ are listed in the equation but definitions are not given.

·        Line 337: “(C-M)” was previously defined as “(CM)”, please select one abbreviation for this term and use it consistently throughout. Check the later instances of this abbreviation too.

·        Line 389: I presume that this should say 0.1x PBS buffer, instead of “PBS buffer”. Saying “PBS buffer” is misleading, as it may lead the reader to think 1x PBS is now being used.

Author Response

Dear Reviewer,

Reviewer 3 Report

Authors describe a DEP-based microfluidic chip and use SERS based detection for cells. Authors develop a COMSOL model and validate the experimental setup using MFC7 and MDA-MD-231 cells. While this paper is interesting and novel, there are some major issues to be resolved before final acceptance. Please consider following-

1) Line 107-111 : The second sentences is redundant given it is repeat of the first one.

2) How is DEP electrode with a 5um tip diameter is fabricated? There is no information about this in the main text. 

3) The MCF-7 and MDA-MD-231 are both adherent cells. For the duration of the experiments in the paper, the cells will settle on the SERS substrate based on gravity. 

Can authors comment on the specificity of DEP capture vs normal gravity ? This is the major issue with this paper because there is no control experiment. 

4) What is cell seeding density ? 

Author Response

Dear Reviewer,

Authors

Round 2

Reviewer 3 Report

The author response to my comments is acceptable. 

Author Response

We would like to thank the Reviewer one more time for the valuable comments.